

# Microbiome patterns across the gastrointestinal tract of the rabbitfish *Siganus fuscescens*

Shaun Nielsen, Jackson Wilkes Walburn, Adriana Vergés, Torsten Thomas and Suhelen Egan

Centre for Marine Bio-Innovation and School of Biological Earth and Environmental Sciences, University of New South Wales, Sydney, Australia

## ABSTRACT

Most of our knowledge regarding the biodiversity of gut microbes comes from terrestrial organisms or marine species of economic value, with less emphasis on ecologically important species. Here we investigate the bacterial composition associated with the gut of *Siganus fuscescens*, a rabbitfish that plays an important ecological role in coastal ecosystems by consuming seaweeds. Members of Firmicutes, Bacteroidetes and delta-Proteobacteria were among the dominant taxa across samples taken from the contents and the walls (sites) of the midgut and hindgut (location). Despite the high variability among individual fish, we observed statistically significant differences in beta-diversity between gut sites and gut locations. Some bacterial taxa low in abundance in the midgut content (e.g., *Desulfovibrio*) were found in greater abundances on the midgut wall and within the hindgut, suggesting that the gut may select for specific groups of environmental and/or food-associated microorganisms. In contrast, some distinct taxa present in the midgut content (e.g., *Synechococcus*) were noticeably reduced in the midgut wall and hindgut, and are thus likely to be representative of transient microbiota. This is the first assessment of the bacterial diversity associated with the gut of *S. fuscescens* and highlights the need to consider the variability across different gut locations and sites when analyzing fish gut microbiomes.

## INTRODUCTION

Microbial symbioses are key to the survival of multicellular organisms with individuals capable of harboring diverse communities of beneficial microorganisms. In particular, gut bacteria play a vital role not only in host nutrition, but also in mediating host immune functions, host development and even influencing host behavior (*Collins & Bercik, 2009*; *Hansen et al., 2012*; *Greenhalgh et al., 2016*). Recent studies have highlighted the diversity and role of gut microbiomes in terrestrial animals, with a particular focus on humans (see reviews by *Hacquard et al., 2015*; *Colston & Jackson, 2016*; *Greenhalgh et al., 2016*). In contrast, relatively little is known about the gut microbiota of marine vertebrates, with most studies focusing on species of economic value (*Colston & Jackson, 2016*).

Corresponding author
Suhelen Egan, s.egan@unsw.edu.au

Fish are the most diverse group of vertebrates in the marine environment. They occupy a wide range of habitats, have highly varying diets and often play important trophic and ecological roles. While our understanding of the role and characteristics of fish gut microbiota is still very limited, studies to date indicate that the bacterial composition of the fish gastrointestinal (GI) track is species-specific and influenced by host physiology as well as environmental conditions, such as diet, salinity and geographic location (*Sullam et al., 2012*; *Wong & Rawls, 2012*; *Clements et al., 2014*; *Romero, Ringø & Merrifield, 2014*; *Givens et al., 2015*).

Diet is arguably one of the strongest influences on gut microbiotia and studies in mammals suggest a trend of increasing bacterial diversity from carnivores to omnivores to herbivores (*Ley et al., 2008*; *David et al., 2014*). Given the key role of microbial fermentation in the conversion of algal biomass into absorbable short chain fatty acids (SCFA), this trend of increased or distinct bacterial diversity in herbivores is likely to also hold true for fish. A meta-analysis of the bacterial communities associated with fish suggested that herbivorous species have a core microbiome consisting of members from the order Clostridiales, Bacteroidales and Verrucomicrobiales, all of which are known in other gut systems to be important for the digestion of plant-based materials (*Sullam et al., 2012*). *Sullam et al. (2012)* further highlight the resemblance between the composition of the gut microbiota of herbivorous fishes with that of terrestrial mammals suggesting, albeit with caution (*Clements et al., 2014*), that fish may represent the first vertebrate hosts for many of the common gut bacterial taxa (*Wong & Rawls, 2012*).

Herbivorous fishes play a key role in healthy tropical coral reef ecosystems by consuming seaweeds that can otherwise outcompete corals (*McCook, Jompa & Diaz-Pulido, 2001*). They can consume nearly 100% of daily algal primary production in coral reef ecosystems (*Carpenter, 1986*) and studies from around the world have shown that the absence of these herbivores can result in profound shifts in ecological communities, e.g., from coral to seaweed dominance (*Hughes et al., 2007*). Rabbitfish belonging to the genus *Siganus* are one of the most important consumers of macroalgae in Indo-Pacific coral reefs (*Bennett & Bellwood, 2011*; *Michael et al., 2013*; *Gilby, Tibbetts & Stevens, in press*). Recently, some range-expanding warm-water rabbitfish have also been implicated in the overgrazing of temperate canopy seaweeds around the world, leading to profound shifts in ecological communities from algal forests to turf-dominated systems (*Vergés et al., 2014*; *Bennett et al., 2015*; *Vergés et al., 2016*). Studies of *Siganus stellatus* show that this species harbors a gut microbial community typical for marine herbivores (*Miyake, Ngugi & Stingl, 2015*). However, the majority of fish microbiome studies such as this one have been performed on faeces or samples of the entire GI tract (i.e., combined content and wall or combined gut regions), thus making the distinction between transient (i.e., allochthonous) food-associated microbiota and the potentially true autochthonous microbiota a challenge. Furthermore, studies on insects (e.g., termite) and mammals (e.g., humans) have shown that distinct section of the gut support different microbial communities that are driven by distinct metabolic processes (*Brune & Dietrich, 2015*; *Jandhyala et al., 2015*). To what extent fish guts have a similar compartmentalization is however poorly understood.

The aim of the present study was to gather baseline data on the microbial communities associated with the gut of *S. fuscescens*, a tropical/subtropical seaweed-consumer that
is linked to the loss of kelp forests in eastern and western Australia as well as in Japan (*Yamaguchi, 2010*; *Bennett et al., 2015*; *Vergés et al., 2016*). We examined microbial communities among individuals and compared communities at different gut locations (i.e., midgut and hindgut) and sites within gut locations (i.e., content and wall). This design allowed us to determine if distinct bacterial taxa are enriched in the different gut locations (i.e., midgut and hindgut), if microbial communities in the gut content (lumen) differed from those associated directly with the gut wall (mucosa), and how the relationship between communities within the content and wall changes along the gut.

## METHODS

### Sampling and DNA extraction

*Siganus fuscescens* individuals were collected by spearfishing from One Tree Island (23.5076°S, 152.0916°E), in the Southern Great Barrier Reef ($n = 4$) with permission from the Great Barrier Reef Marine Park (permit G14/36866.10) and the Queensland Fisheries Department (permit 170194) and with full University of New South Wales animal ethics approval (permit 13/29A). Fish were collected from coral and sand dominated habitats in lagoon or shallow outer reef habitats. Fish were transported in ice to the laboratory and dissected within 3 h to separate the entire gut contents, which were stored at −80 °C until further processing. No fish with punctured intestinal tracts were used in the analyses. From each fish, the intestinal tract was further dissected using a sterile scalpel to separate the midgut (immediately after the stomach) and the hindgut (immediately before the anus) sections. For each of these two sections, the gut content was squeezed out and the gut wall was separated out. Gut wall samples were washed twice with sterile artificial seawater (ASW) to remove any remaining gut content. Total DNA was extracted from gut content and gut walls using the PowerSoil® DNA isolation kit (Mo Bio, San Diego, CA, USA) according to the manufacturers instructions and thereafter stored at −20 °C.

### 16S rRNA gene sequencing and analysis

The 16S rRNA gene was amplified by PCR from total DNA using the method outlined by *Lundberg et al. (2013)*. Briefly, the variable region V4 was targeted with primers 515F and 806R with the PCR including a peptide nucleic acid (PNA) clamp (5′-GGCTCAACCC TGGACAG-3) that suppresses amplification of plastid DNA. PCR products were pooled and sequenced on a MiSeq platform with 2 × 250 bp chemistry at the Ramaciotti Centre for Genomics (UNSW). Paired end sequences were merged into contigs, quality filtered, taxonomically classified and clustered into operational taxonomic units (OTUs) using MOTHUR (*Schloss et al., 2009*) and the associated MiSeq pipeline (*Kozich et al., 2013*), but with minor changes. Briefly, singleton contigs were removed after the pre-clustering step, and were classified using the GreenGenes taxonomic outlines (*DeSantis et al., 2006*) with 60% confidence threshold.

### Design and statistical analysis

The experimental design we used to examine the bacterial communities in the guts of *S. fuscescens* had three factors including fish individuals ('Fish'-four levels), gut location

('GutLoc'-two levels, migut vs hindgut), site ('Site'-two levels, content vs. wall) and the interaction between gut location and site (GutLoc: site-4 levels, midgut:content, midgut:wall, hindgut:content, hindgut:wall). We treated Fish as a fixed blocking factor over a random one, since it has been suggested that random factors should have at least five or more levels for efficient estimation of variance parameters (*Bates, 2010*; *Zuur, Hilbe & Ieno, 2013*).

We compared bacterial communities in terms of alpha and beta diversity, and further looked for differentially abundant taxa within communities. For the diversity measures, we first randomly subsampled (rarefied) each sample to a total of 10,000 counts to account for uneven sequencing depth among the samples. We conducted this procedure 500 times and took the average to reduce randomisation effects on our subsampled data. The non-subsampled data was used for detecting differentially abundant taxa, as the modeling methods used here (generalised linear models, GLMs) can take effects of sequencing depth into account (using an offset term).

The observed number of OTUs and the Shannon diversity coefficient were used as alpha diversity measures for species richness and diversity, respectively. Linear models were constructed and analysis of variance (ANOVA) was used to test for significance of model terms. The Bray–Curtis dissimilarity was used as a beta-diversity measure after square root transformation of relative abundances. Dissimilarities were visualized using Principle Coordinate Analysis (PCoA), and we further used Canonical Analysis of Principal coordinates (CAP) (*Anderson & Willis, 2003*) to visualize dissimilarities constrained on each fixed factor (except Fish) within the experimental design (Gut Location, Gut Site and their interaction) and conditional on the Fish factor (i.e., removal of the effect of Fish before visualizing the other factors). Permutational Multivariate Analysis of Variance (PERMANOVA) was used for hypothesis testing using 999 permutations of the data at hand. Negative binomial GLMs were constructed to detect differentially abundant taxa, as a strong mean–variance relationship of OTU counts was observed. Models were created by taking into account the total number of counts per sample as an offset in the GLM, with the response variable the expected count of an OTU given the sequencing depth. P values were calculated using 999 bootstraps of residuals (resampling rows of the data). Models and P values were generated using the R package MVAbund (*Wang et al., 2012*). OTUs were considered to significantly differ between treatments if the likelihood of the observed test statistics was <5% ($P < 0.05$).

## RESULTS AND DISCUSSION

The mottled spinefoot rabbitfish (*Siganus fuscescens*) is a common herbivorous fish found in Indo-Pacific tropical reefs that plays an important role in the control of algal growth on coral reef systems (*Bennett & Bellwood, 2011*; *Michael et al., 2013*; *Gilby, Tibbetts & Stevens, in press*) yet to date nothing is known about its gut microbiome. Here we investigated the bacterial community in the gut contents and in direct association with the gut wall from individual adults. Bacterial communities were assessed via 16S rRNA gene amplicon sequencing using the Illumina MiSeq platform. Prior to DNA extraction, a high degree of algal material was observed in the gut samples (data not shown), thus samples were amplified with a

PNA-clamp designed to bind to plastid DNA (*Lundberg et al., 2013*). After quality filtering, multiple sequence alignment and clustering at 97% identity, a total of 1,220 OTUs were detected across all 16 samples.

Despite the use of the PNA-clamp, there were 25 OTUs (from a total of 1,220) classified as chloroplast and these were removed prior to downstream analysis. Rarefaction curves indicated that each sample was sequenced nearly to saturation (Fig. S1A) and good coverage was still achieved when the total sequencing depth of each sample was subsampled to 10,000 (Fig. S1B).

We examined alpha diversity within the gut microbiome of *S. fuscescens* using the number of observed OTUs (species richness, Fig. S2) and the Shannon-Weaver index (species diversity, Fig. 1). Both metrics showed similar patterns, including variation among individual fish, and greater alpha diversity on the gut wall (mucosa) relative to the gut content (lumen) regardless of gut location (Tables S1A and S1B). While differences in alpha diversity between wall and gut content was generally smaller in the hindgut than in the midgut, the large within group variance and small sample size resulted in limited statistical support for an interaction (Tables S1A and S1B). Interestingly, an individual fish (individual A) showed extreme variation in species diversity, which was due to a greater evenness in the OTU abundance distribution within the midgut of this fish and which was unlike the trends observed in every other sample (Fig. 2).

Analysis of beta diversity using the Bray–Curtis dissimilarity coefficient revealed community dissimilarities ranged between 88.9 and 22.9%. With the exception of the individual A, there was no discernible pattern in the clustering of samples between fish (Fig. 3A). Rather, samples tended to cluster according to the gut location (i.e., midgut or hindgut, $F_{1,9} = 5.67$, $P < 0.001$, Fig. 3B) or gut site (i.e., gut wall or gut content, $F_{1,9} = 2.72$, $P = 0.011$, Fig. 3C). While bacterial communities of the hindgut content and wall appeared more similar to each other, and together different from the variable communities in the midgut (Fig. 3D), there was not enough statistical power to support this observation given the small sample size ($F_{1,9} = 1.49$, $P = 0.133$, Table S1C). It is important to note that while care was taken to avoid sampling bias, it is possible that the high variability between samples may partly be explained by technical variability during sample collection (e.g., slight differences in the removal of content or rinsing of the gut walls) in addition to true biological variation.

Differences in microbiome composition between gut regions and sites (i.e., mucosa and lumen) have previously been reported for Atlantic salmon (*Salmo salar*) (*Gajardo et al., 2016*). However, in contrast to our observations with *S. fuscescens*, overall less variation between individual salmon were reported. This finding may relate to general differences between the gut microbiota of herbivores and omnivores (*Ley et al., 2008*), but could also reflect the defined diet of captive salmon compared to the natural and likely more variable diet of the *S. fuscescens*. In line with this hypothesis, we found that the variability among fish could be best explained by differences in the microbial community associated with the midgut content rather than differences between other gut locations or sites (Fig. 3). While *S. fuscescens* is known to consume algae, it feeds on both adult macrophytes and the epilithic algal matrix (*Wilson et al., 2003*), which contains a nutritionally disparate mix of filamentous algae, organic matter, detritus and sand. Further, closely related species are also

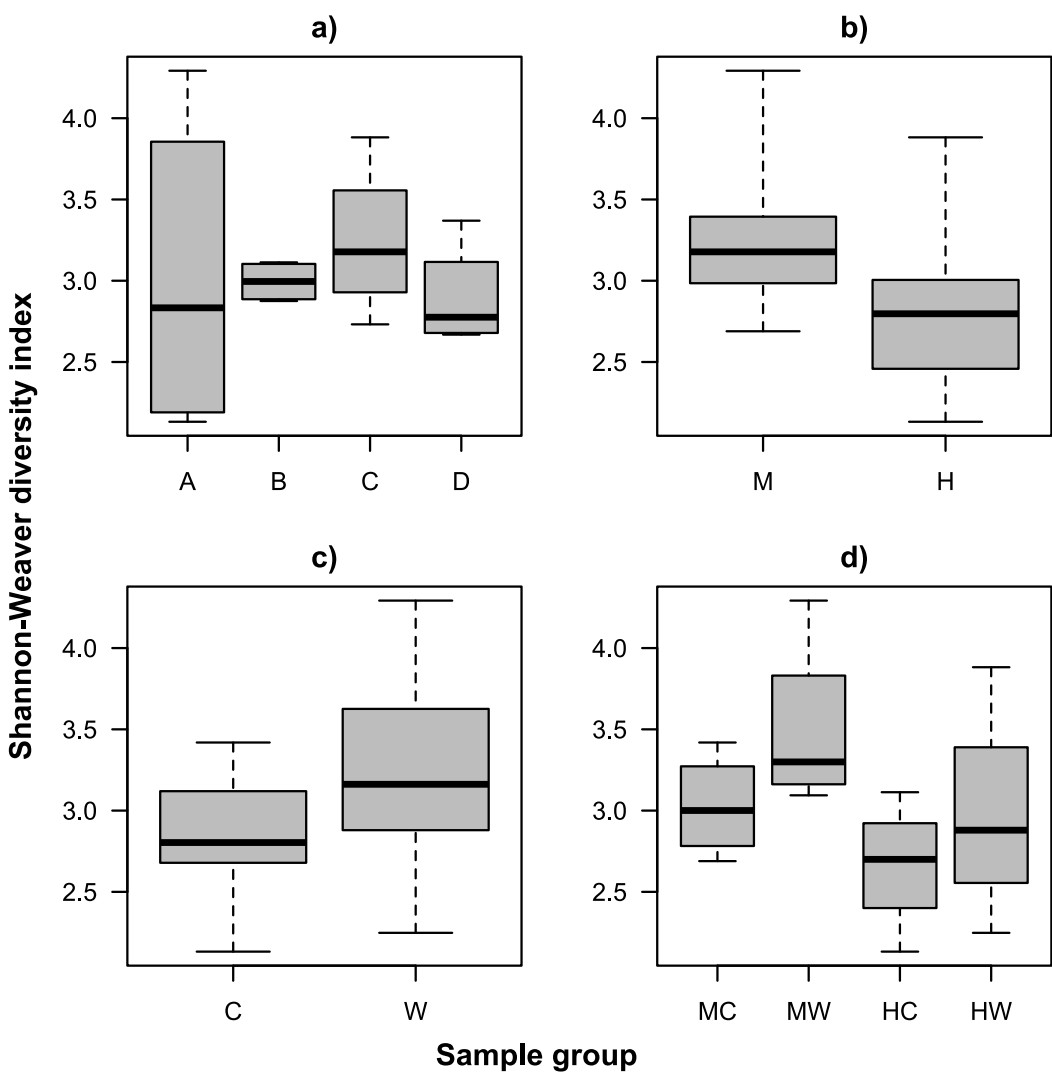

**Figure 1** Bacterial species diversity (Shannon-Weaver diversity index) within the gut of the mottled spinefoot rabbitfish (*Siganus fuscescens*) among (A) individual fish ($n = 4$), (B) gut locations (M, mid; H, hind), (C) gut sites (C, content; W, wall) and (D) gut sites within gut locations (MC, mid-content; MW, mid-wall; HC, hind-content; HW, hind-wall), given a total sampling depth of 10,000 16S rRNA gene counts per sample.

known to be opportunistic omnivores, occasionally feeding on invertebrate animals, such as jellyfish (*Bos, Cruz-Rivera & Sanad, 2016*). Thus, the high variation in the communities associated with the midgut contents of individual fish could be explained by variation in the food material consumed by the fish prior to collection. These results further highlight the value of sampling from discrete gut regions when assessing the bacterial community of these animals.

The observation that bacterial communities can vary within the midgut, but tend to become similar to one another towards the hindgut, suggests that specific gut locations select for specific bacterial groups. To determine which bacterial taxa were represented in each of the gut locations and sites, OTUs were taxonomically classified using the GreenGenes

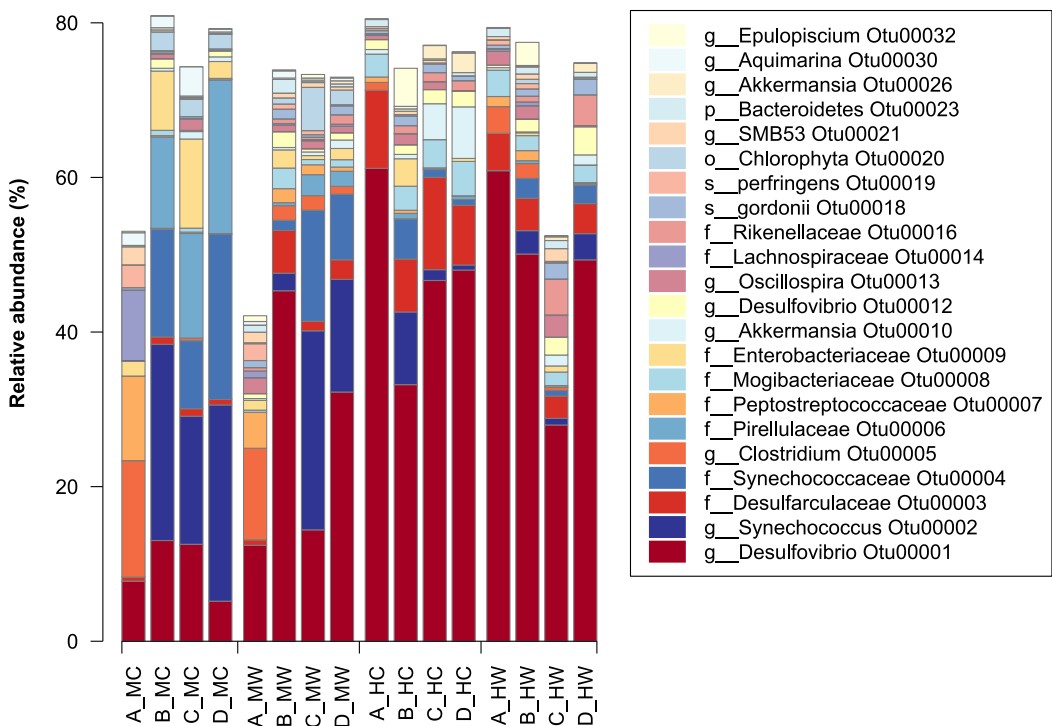

**Figure 2** **Relative abundance of bacterial OTUs in the within the gut of the mottled spinefoot rabbit-fish (*Siganus fuscescens*).** Only OTUs with relative abundances >1% are shown, and are described by the lowest taxonomic assignment. Horizontal axis labels represent Fish individual (A–D) followed by an underscore and then gut sites within gut locations (MC, mid-content; MW, mid-wall; HC, hind-content; HW, hind-wall).

taxonomic outlines (*DeSantis et al., 2006*). In total 25 phyla were observed, however only six of these were found in relative abundances >0.5% (Fig. S3). Overall delta-Proteobacteria, Firmicutes and Cyanobacteria made up a large proportion of the gut microbiota, and there was clear variation of these taxa across the different gut locations and sites. For example, the Cyanobacteria were found largely within the midgut content, while the delta-Proteobacteria were largely found in the distal gut regardless of the site sampled.

At the individual OTU level, only 23 of the total OTU's observed had a relative abundance >1% (Fig. 2). The most dominant OTU (OTU00001) belonged to the genus *Desulfovibrio* (delta-Proteobacteria), which in some samples constituted as much at 60% of the relative abundance in the community. This OTU was present in all samples, but had a low abundance in the midgut, especially within the content, and a high abundance in the distal gut (Fig. 2). In contrast, the second most abundant OTU (OTU00002), assigned to the genus *Synechococcus*, showed the greatest abundance within the midgut content, but was significantly lower within the distal region (Fig. 2). Notably, other closely related OTUs had similar patterns in abundance to OTU00001 and OTU00002 with, for example, members of the family *Desulfovibrionaceae* being in lower abundance within the midgut content and in greater abundance within the midgut wall and the distal gut (Fig. S4). In contrast, members of the family *Synechococcaceae* were only abundant in the midgut content (Fig. S4) and

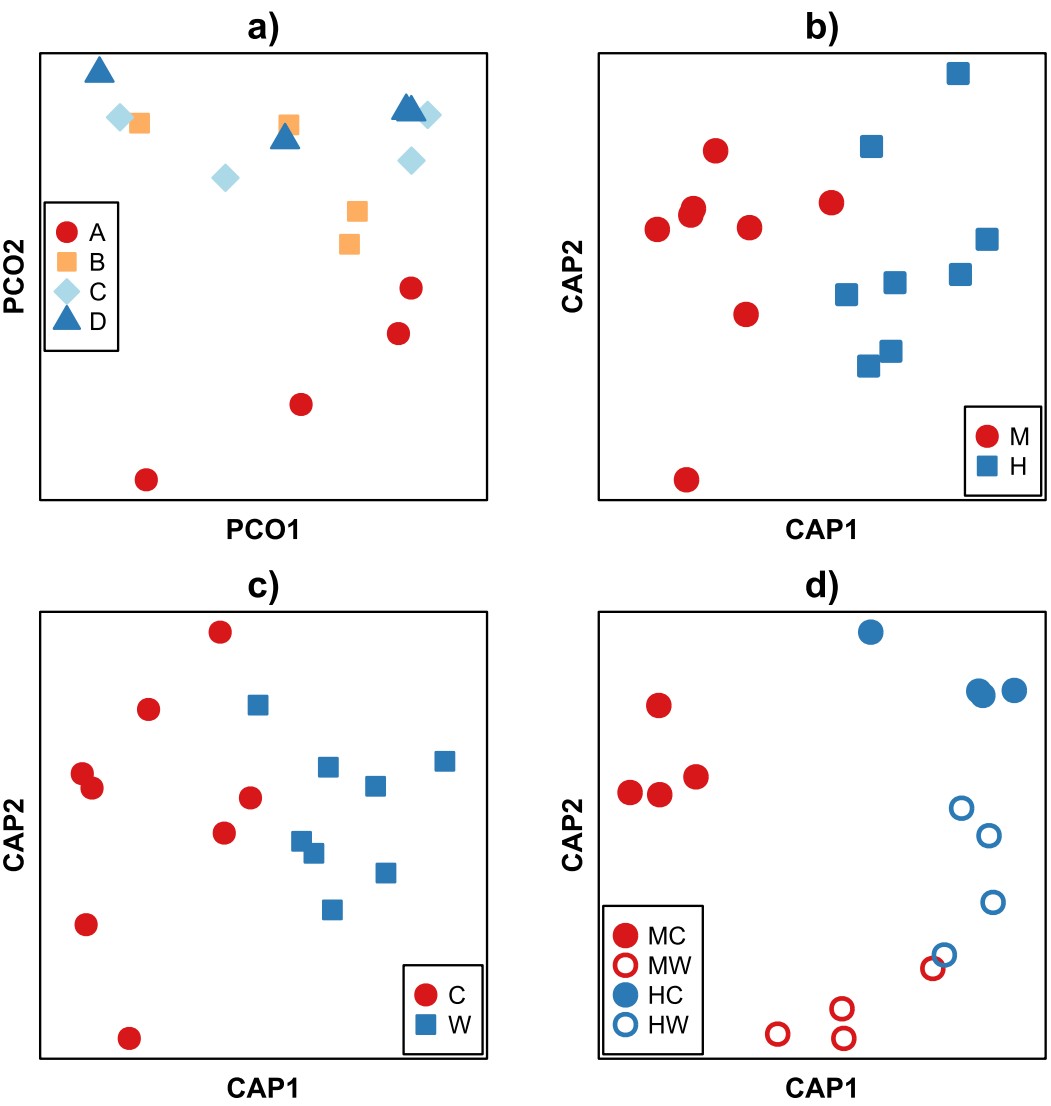

**Figure 3** Ordination of Bray–Curtis dissimilarities between bacterial communities within the gut of the mottled spinefoot rabbitfish (*Siganus fuscescens*). Ordination of Bray-Curtis dissimilarities between bacterial communities within the gut of the mottled spinefoot rabbitfish (*Siganus fuscescens*) compared among (A) individual fish (*n* = 4) using Principle Coordinate Analysis (PCoA), and among (B) gut locations (M, mid; H, hind), (C) gut sites (C, content; W, wall) and (D) gut sites within gut locations (MC, mid-content; MW, mid-wall; HC, hind-content; HW, hind- all) after conditioning on individual fish using Canonical Analysis of Principal coordinates (CAP). Relative abundances, given a total sampling depth of 10,000 16S rRNA gene counts per sample, were square root transformed before calculation of dissimilarities. Variance explained in (A) for the horizontal axis = 34 % and vertical axis = 23 %.

thus are likely to be only transient members of these communities. This suggests that specific bacterial groups abundant within the marine environment (e.g., *Synechococcaceae*) are largely lost thereafter, while other groups (e.g., *Desulfovibrionaceae*) can persist, and flourish, while travelling through the gut.

Given the qualitative observations above, we then modeled OTU counts to detect the number of individual OTUs that were differentially abundant across individuals and within

specific sites and gut locations (Figs. 4A–4B). We found that 39 OTUs varied in abundance across individual fish (Table S2A), while 77 OTUs showed abundance changes across gut locations irrespective of whether we considered the gut wall or the gut contents (Table S2B). A total of 59 OTUs differed across sites irrespective of gut locations (Table S2C) and 50 OTU had abundances that varied depending on both gut locations and sites (Table S2D). Interestingly, while approximately half of the bacteria alternated in abundance between the midgut and hindgut locations (Fig. 4B), differences in abundance between gut wall and wall content were largely a result of the majority of OTUs having greater abundances in the wall relative to the content (Fig. 4C). Furthermore, OTUs that varied depending on both gut locations and sites tended to have low abundances in the midgut content, but similar abundances among the other gut locations and sites (Fig. 4D).

OTU's enriched in the gut wall belonged to the order Clostridiales and Bacteroidales, including several taxa commonly found in the GI tract of animals, such as *Clostridium* species and members of the family *Rikenellaceae* (*Rajilić-Stojanović & De Vos, 2014*; *Mao et al., 2015*). These bacterial groups are efficient fermenters of plant and algal material and thus their close association with the intestinal mucosa presumably ensures direct uptake of fermentation products (i.e., SCFA) to the host bloodstream (*Clements et al., 2014*; *Rajilić-Stojanović & De Vos, 2014*). OTUs that were in greater abundance in the midgut wall, and either site of the hindgut relative to the midgut content were assigned to the families *Desulfarculaceae*, *Rikenellaceae* and *Ruminococcaceae*. Contrasting patterns were observed for OTU's in the family *Rhodobacteraceae* and genera *Rubritalea* (Verrucomicrobiaceae). These bacterial taxa are ubiquitous in the marine environment (*Brinkhoff, Giebel & Simon, 2008*; *Hedlund, Yoon & Kasai, 2015*) and their greater abundance in the midgut contents compared to the gut wall and hindgut regions suggests that similar to the cyanobacteria, they may represent ingested allochthonous members of the *S. fuscescens* gut microbiome rather than true gut symbionts. OTUs that increased in abundance between the midgut and hindgut (>0.5% change) were assigned to the genera *Desulfovibrio*, *Akkermansia*, *Treponema* and the family *Mogibacteriaceae*. In contrast, OTUs assigned to the genera *Aquimarina*, *Acinetobacter*, and the families *Enterobacteriaceae*, *Peptostreptococcaceae* and *Pirellulaceae* decreased in abundance.

Overall the observed OTU abundance patterns are likely to reflect changes in the environmental parameters between the gut locations and sites. The enrichment of strictly anaerobic *Clostridium* species and sulfate reducing bacteria (SRB), such as *Desulfovibrio* in the hindgut regions implies an anoxic and sulfate-rich environment, which shows a clear compartmentalization of microbial communities across the gut axis driven by host/environmental factors. The abundance of SRB further implies that fermentation of algal material by, for example, *Clostridium* spp. is likely to be coupled predominately with sulfate reduction rather than acetogenesis or methanogenesis, in line with studies that demonstrated higher rates for sulfate reduction than methanogenesis in the marine herbivorous fish (*Mountfort, Campbell & Clements, 2002*). The dominance of SRB has also been seen for other marine herbivores (*Hong et al., 2011*), including *Siganus stellatus* collected from the Red Sea (*Miyake, Ngugi & Stingl, 2015*), and may reflect a diet rich in sulfated algal polysaccharides.

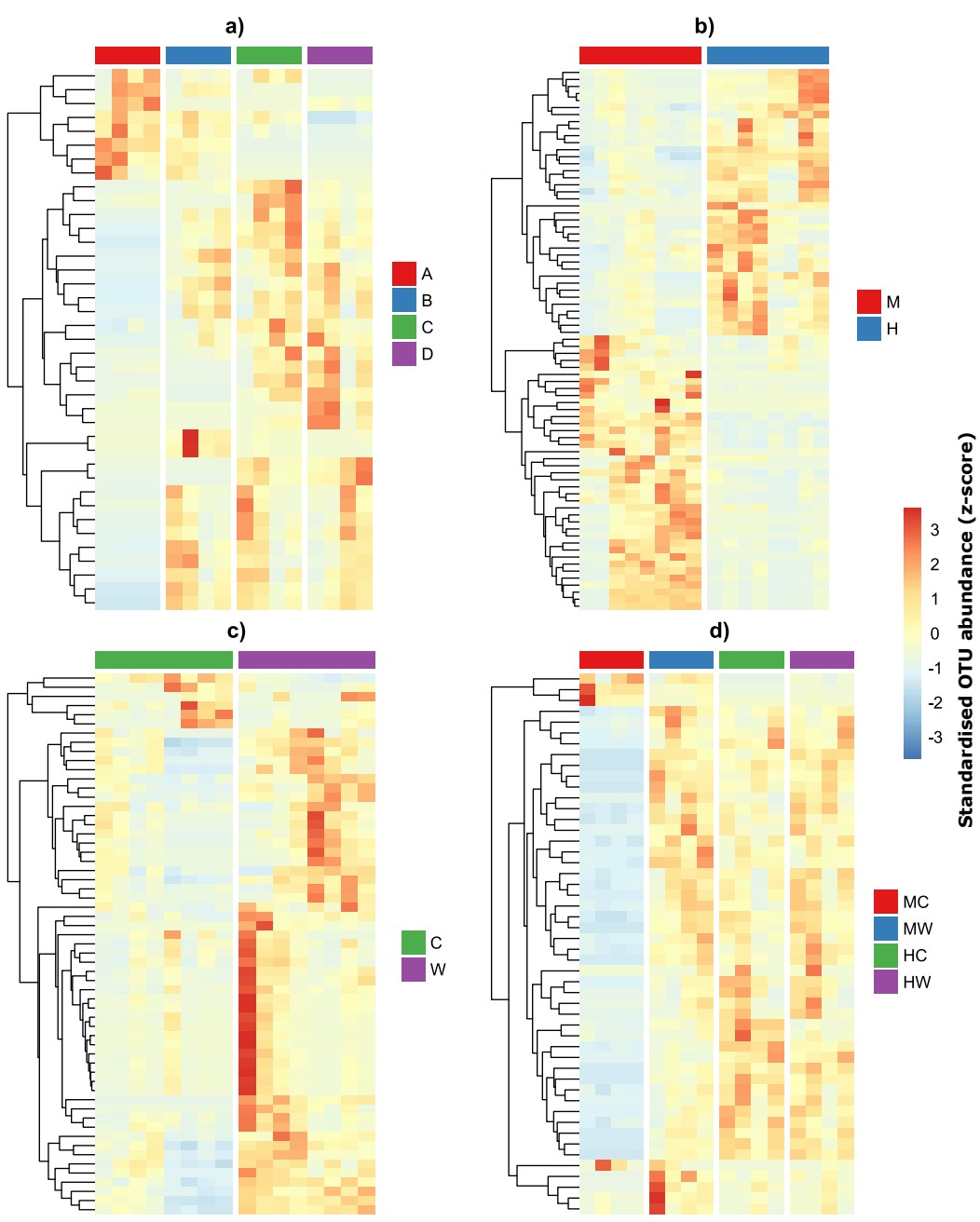

**Figure 4** Differentially abundant OTUs within the gut of the mottled spinefoot rabbitfish (*Siganus fuscescens*) among (A) individual fish (*n* = 4), (B) gut locations (M, mid; H, hind), (C) gut sites (C, content; W, wall) and (D) gut sites within gut locations (MC, mid-content; MW, mid-wall; HC, hind-content; HW, hind-wall). OTU abundances have been *z*-score transformed and thus show the number of standard deviations an OTUs abundance is from the mean abundance of that OTU.

In that regard hindgut processes in herbivorous fish are distinct to those, for example, of termites, which mostly consume low-sulfate organic material and hence couple fermentation to methanogenesis (*Brune & Dietrich, 2015*).

We further suggest marine sediments inadvertently ingested by *Siganus* sp., while it is feeding on small, benthic algae, as an additional source of both sulfate and SRB. Thus for an actively grazing fish, such as *S. fuscescens*, the composition of their gut microbiome and the associated metabolic functions are likely to be heavily determined by both the diet and the microorganisms that they ingest from their environment. Similar processes of environmental acquisition, followed by host enrichment of gut bacteria have been suggested for a number of marine and freshwater fish species (*Nayak, 2010*; *Givens et al., 2015*; *Parris et al., 2016*; *Yan et al., 2016*; *Stephens et al., 2016*). Therefore, in addition to the influence of diet itself (e.g., nutrient content or trophic level), further investigations on the processes that control gut-microbiome assembly in *S. fuscescens* and other marine fish should assess the relative contribution made by diet-associated microorganisms.

In conclusion, despite variation in gut microbial communities across individual *S. fuscescens,* specific bacterial OTUs were significantly enriched within different locations and sites of the gut. The microbial community associated with samples taken from midgut content was more variable than the other gut samples and contained taxa (e.g., cyanobacteria) that are likely to be associated with what is being ingested with the food, rather than being true members of the gut microbiome. In contrast, the midgut wall, hindgut wall and the hindgut content all appear to support a less transient microbiome, with a higher abundance of taxa known to play an important role in fermentation e.g., Firmicutes and Bacteroidetes. These observations highlight the need for future studies to take into account not only the spatial distribution of bacterial groups along the gastrointestinal tract, but also the relative contribution of environmentally acquired microorganisms to the gut microbiome of ecologically important marine fish.

## ACKNOWLEDGEMENTS

We are thankful to Derrick Cruz for his help collecting the fish samples.

### Funding

This work was supported by the Centre for Marine BioInnovation, UNSW Sydney and by the Sea World Research and Rescue 343 Foundation (grant SWR/1/2014 to A Vergés). There was no additional external funding received for this study. The funders had no role in study design, data collection and analysis, decision to publish, or preparation of the manuscript.

### Grant Disclosures

The following grant information was disclosed by the authors:
Centre for Marine BioInnovation.

UNSW Sydney.

Sea World Research and Rescue 343 Foundation: SWR/1/2014.

## Competing Interests

Adriana Vergés and Torsten Thomas are Academic Editors for PeerJ.

## Author Contributions

- Shaun Nielsen performed the experiments, analyzed the data, wrote the paper, prepared figures and/or tables, reviewed drafts of the paper.
- Jackson Wilkes Walburn performed the experiments, analyzed the data, reviewed drafts of the paper.
- Adriana Vergés and Torsten Thomas conceived and designed the experiments, contributed reagents/materials/analysis tools, wrote the paper, reviewed drafts of the paper.
- Suhelen Egan conceived and designed the experiments, analyzed the data, contributed reagents/materials/analysis tools, wrote the paper, prepared figures and/or tables, reviewed drafts of the paper.

## Animal Ethics

The following information was supplied relating to ethical approvals (i.e., approving body and any reference numbers):

This work was performed with full ethics approval from the University of New South Wales (ethics permit 13/29A).

## Field Study Permissions

The following information was supplied relating to field study approvals (i.e., approving body and any reference numbers):

This research was performed under permit G14/36866.1 from the Great Barrier Reef Marine Park and permit 170194 from the Queensland Fisheries Department.

## Data Availability

Raw sequencing data is deposited in the NCBI Short Read Archive (SRA) under accession number PRJNA356981.

## Supplemental Information

Supplemental information for this article can be found online at http://dx.doi.org/10.7717/peerj.3317#supplemental-information.

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
