# Peer review of "Microbiome patterns across the gastrointestinal tract of the rabbitfish Siganus fuscescens"

_PeerJ, doi:10.7717/peerj.3317_

## Round 0.1 · original submission · Major Revisions

Dear authors,

Your paper requires moderate/major revision. Please provide a detailed point-by-point response to all of the reviewers' comments along with your revised manuscript.

Reviewer 1 ·

Basic reporting

Adequate

Experimental design

Overall, the manuscript is rather simple. Of course a large volume of work has been conducted and at a pretty high experimental level. A small sampling size prevented authors from making more substantial conclusions.

Validity of the findings

The claim (line 21 -23) that “ … clear differences in bacterial alpha and beta diversity were observed across both gut locations … and gut content and gut walls” is incorrect. According, to box plots (Figure 1) Shannon diversity indexes are not statistically dissimilar for each group of variables. According to Figure 2, gut walls and gut lumen communities are also not dissimilar.
Can be fixed though: see the comments for the authors

Additional comments

Indicate that a small sample size prevented making any statistical comparisons of the communities and focus on the description of individual dominant OTUs - exactly how it was done in the manuscript.

Reviewer 2 ·

Basic reporting

no comment

Experimental design

no comment

Validity of the findings

no comment

Additional comments

The manuscript describes a thorough 16S-rDNA-based analysis of bacterial community structure associated with intestinal tract of an abundant herbivorous rabbitfish Siganus fuscescens. The study is well-designed, and goes refreshingly beyond the routine comparison of overall gut community structure, by looking at bacterial communities associated with different broad microhabitats along the gut axis, such as different gut regions (midgut and the hindgut), as well as, those associated with the gut wall and lumen. Because of the attention to microhabitat-associated communities in the gut and the detailed discussion of potential biochemical processes that would explain community composition, I believe this study contributes to a basic understanding of intestinal symbioses.

The experiments are well-designed, the analysis is thorough, and the manuscript is well written. I have a few comments (all minor) that I would like addressed in a revision of the manuscript.

1. Figure 2 has a lot of redundancy. A careful choice of symbols and colours should considerably simplify the 4-panel figure.

2. As admitted in the manuscript, the authors’ dataset does not show a clear clustering of samples by microhabitat. This could indicate considerable biological variability among individual fish , which has been addressed by the authors, but I am not satisfied by their explanation of potential biases introduced by technical variability. I would suggest an elaboration on how preparation of the samples (e.g., differences in squeezing luminal contents for separating the wall samples from contents).

3. Given that the sampling of different gut regions was an important novelty of the study, I was missing mention of how spatial, physicochemical, and metabolic parameters could drive gut community structure in complex gut communities, which has been thoroughly studied in mammals and termites. A comparison to such more familiar model systems (in the introduction or the discussion) could help readers understand the premise and highlight the novelty of the authors’ study.

4. Miscellaneous comments:
Ln 22 (and elsewhere): Spell out midgut and hindgut for clarity.
Ln 59--60: phyla -> orders
Ln 86: Western -> western
Ln 155: OTU -> OTUs
Ln 289: Zac Stephens et al. -> Stephens et al.
Ln 291: Given that Miyake et al. (2015) have already shown a strong influence of diet on the gut microbiota of marine fish, what do the authors mean when they suggest that the studies of gut community assembly should also assess “the influence of diet and diet-associated microorganisms”? And what is a “diet-associated microorganism”?
Ln 293--302: I enjoyed reading the entire manuscript, except the concluding paragraph, which seems really rushed and unclear, e.g., “The wall and the hindgut appears to represent a less transient microbiome” - wall of the midgut or the hindgut? In order to be conclusive, this paragraph needs to be reworked, and the authors must clearly state what future directions they are trying to indicate.

---

## Round 0.2 · accepted · Accept

Thank you for your submission.

Reviewer 2 ·

Basic reporting

no comment

Experimental design

no comment

Validity of the findings

no comment

Additional comments

Nielsen and colleagues have nicely revised their manuscript and have satisfactorily addressed all my concerns.